# Surgery for Hereditary Diffuse Gastric Cancer: Long-Term Outcomes

**DOI:** 10.3390/cancers14030728

**Published:** 2022-01-30

**Authors:** Joseph D. Forrester, Deshka Foster, James M. Ford, Teri A. Longacre, Uri Ladabaum, Sara Fry, Jeffrey A. Norton

**Affiliations:** 1Department of Surgery, Stanford University Medical Center, Stanford, CA 94305, USA; jdf1@stanford.edu (J.D.F.); dsfoster@stanford.edu (D.F.); sfry@stanfordhealthcare.org (S.F.); 2Departments of Medical Oncology, Stanford University Medical Center, Stanford, CA 94305, USA; jmf@stanford.edu; 3Department of Pathology, Stanford University Medical Center, Stanford, CA 94305, USA; longacre@stanford.edu; 4Department of Gastroenterology, Stanford University Medical Center, Stanford, CA 94305, USA; ladabau@stanford.edu

**Keywords:** hereditary diffuse gastric cancer (HDGC), CDH-1 mutation, gastric cancer

## Abstract

**Simple Summary:**

This study reports the long-term results of total gastrectomy for patients with a family history of gastric cancer and CDH1 gene mutations that predispose to hereditary diffuse gastric cancer (HDGC). Total gastrectomy was performed in 8 symptomatic patients and 22 asymptomatic patients of whom only 3 had HDGC diagnosed preoperatively. 7 of 8 symptomatic had metastatic lymph nodes with cancer. 21 of 22 asymptomatic patients had gastric cancer localized to the stomach and each was cured. 15 of those patients had 9-year follow-up. Each had significant weight loss (23% body weight) with a normal body mass index, 40% had bile reflux controlled with medication, and each returned to work and said they would do it again. Long-term quality of life following gastrectomy was acceptable.

**Abstract:**

Introduction: Gastric cancer is inherited as an autosomal dominant condition in hereditary diffuse gastric cancer (HDGC). The gene associated with HDGC is an E-cadherin gene CDH1. At the time of initiation of this study, it was estimated that 70% of patients who inherited the CDH1 gene mutation would develop gastric cancer. We hypothesized that the rate of signet ring cell cancer in asymptomatic patients with CDH1 mutations may be higher than anticipated and that the surgery could be conducted with acceptable short-term and long-term complications suggesting that the quality of life with the surgery is acceptable. Methods: We prospectively studied the role of total gastrectomy in symptomatic and asymptomatic patients with CDH1 mutations. A total of 43 patients with mutations of the CDH1 gene were studied prospectively, including 8 with symptoms and 35 without symptoms. Total gastrectomy was recommended to each. Quality of life was assessed in patients who underwent prophylactic gastrectomy. Proportions are compared with Fisher’s exact test. Results: In total, 13 (30%) asymptomatic patients declined surgery. Total gastrectomy was performed in 8 symptomatic patients and 22 asymptomatic patients of whom only 3 asymptomatic patients (14%) had endoscopically proven signet ring cell cancer preoperatively, while 21 of 22 (95%) had it on final pathology (*p* = 0.05). Each asymptomatic patient was T1, N0, while seven out of eight symptomatic patients had T3-T4 tumors and six had positive lymph nodes. None had operative complications or operative death. The median follow-up was 7 years. Five (63%) symptomatic patients died, while only one (95%) prophylactic patient died of a non-gastric cancer- or surgery-related issue (*p* = 0.05). A total of 15 prophylactic patients had long-term follow-up. Each had significant weight loss (mean 23%) but all had a normal body mass index. In total, 40% had bile reflux gastritis controlled with sucralfate. Each returned to work and, if given the choice, said that they would undergo the surgery again. Conclusions: Total gastrectomy is indicated for patients who have an inherented CDH1 mutation. Endoscopic screening is not reliable for diagnosing signet ring cell stomach cancer. If patients wait for symptoms, they will have a more advanced disease and significantly reduced survival. Operative complications of prophylactic gastrectomy are minimal, and long-term quality of life is acceptable.

## 1. Introduction

Gastric cancer affects nearly 1 million individuals each year and 70–85% of individuals who develop gastric cancer will die from it. It is the third leading cause of cancer death worldwide [1]. Although most cases of gastric cancer are sporadic, aggregation within families happens in 10% of cases [1]. However, truly hereditary cases of gastric cancer occur in only between 3–5%, the most common form is hereditary diffuse gastric cancer (HDGC) [2]. HDGC was first described in a Maori kindred from New Zealand in 1998 [3]. It is caused by a mutation in the E-cadherin gene CDH1 [1]. Cadherins are a type of cell adhesion molecule important in the formation of adherens junctions to allow cells to adhere to each other [4]. HDGC is inherited as an autosomal dominant trait with a high degree of penetrance [2]. Gene mutation carriers have approximately a 70% lifetime risk of gastric cancer [5,6] and a 23% risk of lobular breast cancer [6]. In 2004, we initiated a prospective study to perform total gastrectomy on patients with CDH1 truncating, but not missense, mutations, irrespective of whether or not they had a preoperatively diagnosed cancer [7]. A major issue in recommending prophylactic total gastrectomy (PTG) to asymptomatic, preoperatively apparently disease-free patients is perceptions of long-term body habitus, energy, weight loss and quality of life after surgery. Outcome data more than two years postoperatively has not been previously reported [1,4,5,6]. We assessed long-term functional outcomes after prophylactic total gastrectomy for CDH-1 mutations in otherwise asymptomatic, imaging-negative and endoscopic-biopsy-negative HDGC patients and compared them to symptomatic patients with HDGC. The hypothesis was that the surgery could be conducted safely with a reasonable long-term quality of life.

## 2. Methods

Beginning in August 2004, all patients with inherited CDH-1 gene truncating mutations were prospectively studied on an IRB-approved clinical protocol [7]. The protocol was concluded in 2018. Patient enrollment has been described previously [7]. Each patient enrolled in this study had a truncating mutation of the CDH1 gene and a family history of HDGC. Preoperative evaluation included a comprehensive physical examination and medical history, laboratory, radiographic and upper endoscopic examinations with high magnification chromoendoscopy with multiple gastric biopsies, endoscopic ultrasonography, computed tomography, and deoxy-glucose PET scan. Regardless of the outcomes of those studies, open total gastrectomy, D-2 lymph node dissection, and Roux-en-Y esophagojejunostomy was recommended to each patient. It was performed after obtaining informed consent during which we estimated that there was a 5% chance of complications (the greatest of which was a leak at the esophago-jejunostomy anastomosis) and a 1–2% chance of death from the surgery. After surgical resection, the stomach and lymph nodes were totally dissected pathologically, as described previously [8]. Demographic, perioperative, histopathologic, clinical, and outcome data were collected. Statistical evaluation of proportions was performed with Fisher’s exact test.

## 3. Results

In total, 43 patients (8 symptomatic, 35 asymptomatic) with a CDH1 gene truncating mutation were studied. Total gastrectomy was recommended to each. Thirteen (30%) patients declined surgery and were advised to get yearly endoscopy with a gastroenterologist close to home. We do not have the results of these studies. For those proceeding with surgery, 8 were symptomatic and 22 were asymptomatic (Table 1). Each symptomatic patient had gastric cancer diagnosed preoperatively on endoscopy and imaging. Three (14%) asymptomatic patients had signet ring cell adenocarcinoma detected during preoperative endoscopy, the remaining 19 patients underwent completely negative preoperative studies as listed in the methods. The 19 patients underwent true prophylactic total gastrectomy and D2 lymph node dissection without a preoperative tissue diagnosis of HDGC and normal imaging. Each prophylactic stomach that was removed appeared absolutely normal at the time of surgery. In the prophylactic surgery group, 18 out of 19 prophylactic (95%) patients had multifocal (mean 13 sites, range 2–23 sites) superficially invasive T1 diffuse signet ring cell adenocarcinoma that was most prevalent in the proximal (cardia) stomach. Each had >20 lymph nodes excised, and none were positive in any patient (N0). One prophylactic patient had no cancer in the specimen (stomach was normal). All patients were maintained on vitamin B12 supplements by monthly injections postoperatively. Surgical margins were negative in each patient. Each symptomatic patient underwent similar surgery and had deeply invasive multifocal T3/T4 signet ring cell adenocarcinoma of the stomach with a Ki-67 labelling of 48–55% (Table 2). No asymptomatic PTG patient had positive lymph nodes, while seven (88%) out of eight symptomatic patients had positive lymph nodes (*p* < 0.0001). Five (63%) out of eight symptomatic patients died from cancer-related issues at 2 years postoperatively, only one prophylactic total gastrectomy patient died from a non-gastric cancer-related issue (*p* = 0.05). There were no postoperative complications, and the median hospital stay was 5 days.

In follow-up, 15 prophylactic gastrectomy patients returned for long-term follow-up (median 9 years; range 5–17 years). Each had significant weight loss (median 23% of preoperative weight, range 7–50%). However, each had a normal body mass index for their age, sex, and height. Each was able to graduate from six meals per day to three meals at an approximate postoperative recovery time of 6–9 months. Eight (57%) patients reported no digestive symptoms after recovery from gastrectomy, while six (43%) reported chronic upper abdominal discomfort associated with bile reflux. Symptoms of bile reflux were well controlled with sucralfate. All patients returned to work and normal activity. None had any evidence of cancer recurrence. Each of the 15 patients stated that they would undergo surgery again despite the presence of side effects in some.

## 4. Discussion

HDGC is the most malignant form of gastric cancer. It is present in multiple locations throughout the entire stomach as signet ring cell adenocarcinoma. It is nearly always in the cardia requiring the surgeon to perform total gastrectomy as the operative procedure of choice. Total gastrectomy is a more challenging operation with a greater potential for life-threatening complications, but there were no operative complications in the current series. While prophylactic mastectomy or thyroidectomy to address genetic risk has become commonplace, prophylactic gastric resection carries a much greater surgical risk in terms of immediate and long-term complications. Yet prophylactic gastrectomy offers a large potential survival advantage because if surgery is delayed until symptoms development, survival is significantly reduced, [2] as is documented in the current study.

Some patients with CDH1 mutations declined surgery. In the current study, 13 patients declined surgery, and this may be because of the planned open operation. Open total gastrectomy may be perceived or experienced as being more painful, which could have influenced these patients’ decisions to not undergo prophylactic surgery. Minimally invasive methods may be more acceptable to patients, particularly those undergoing a prophylactic operation. Increasingly, total gastrectomy is currently being performed by laparoscopic and robotic methods that should further reduce operative side effects, particularly lessening pain in the postoperative period. However, a recent analysis of prospective randomized trials of open versus laparoscopic total gastrectomy in a Cochrane database of 2794 patients in 13 trials show no significant differences in outcomes of laparoscopic vs. open gastrectomy and both are still being performed [9,10].

Since this approach (prophylactic gastrectomy) for the management of HDGC is relatively recent and has a high chance of curing the disease, the long-term sequela and complications of the procedure should be documented and are novel. Further, since this patient population is younger, asymptomatic, and is projected to have excellent survival, the long-term side effects are important to know before recommending invasive surgery. There are three prior studies assessing 2-year results of similar patient populations [11,12,13]. One prior study had a high complication rate requiring additional intervention in 27% of patients and one death (operative mortality 2.5%). Weight loss was measured at 15% and stabilized at 12 months postoperatively [11]. A second study of 32 patients who underwent prophylactic total gastrectomy for CDH1 mutation showed a slow recovery of cognitive function over 9 months. These patients had difficulty eating, with a poor body image in 44% [12]. Finally, another study showed that health-related quality of life declined significantly after total gastrectomy, improved at 6–12 months postoperatively, and then decreased again at 24 months [13]. In two other studies evaluating quality of life in a follow-up of total gastrectomy at two years, more than 60% of patients reported persistent reflux [5,6]. In our extended follow-up, this percentage decreased slightly (40%), suggesting the over time these symptoms improve through lifestyle modifications, pathophysiologic adaption to surgery, or both. However, once we realized that this may be a worrisome common complication, we increased the length of the Roux limb that connected to the esophagus from 20 cm to 30 cm. This may have resulted in less reflux of bile and is our current practice. Prior short-term follow-up studies reported weight loss of 18% [5]. Compared to the median 23% weight loss seen in our study, weight loss in post-gastrectomy patients appears to stop and weight becomes stable over a long-term follow-up. As mild symptoms persist whether the operation is performed open or by less invasive methods, these patients need careful preoperative counseling, collaboration with nutritionists, and close long-term follow-up. Multi-disciplinary care plans are needed for effective management of this unique patient population.

This study has some limitations. First, 30 is a small number of patients studied. This can be explained because HDGC accounts for only 3–5% of gastric cancer. We were one of the first groups to perform surgery for CDH1 mutations, and we have accrued only 30 patients over 17 years. Second, we do not have follow-up on the 13 patients who declined surgery. We do not know if they were found on endoscopy to have signet ring cell cancer or if they have developed worrisome symptoms consistent with HDGC. Third, we do not have a control group who were randomized to undergo repeated endoscopy to see if we can use that method to detect signet ring cell cancer of the stomach early and only perform surgery when a curable cancer is detected preoperatively. We did not include this group because our bias was that gastrectomy was the best approach. Because we had previously developed a method to serial section the entire resected stomach to find signet ring cell cancer [2], we found a higher prevalence of multifocal signet ring cell gastric cancer than expected. We expected only 70% of patients to have signet ring cell adenocarcinoma in the removed stomach [1], but 95% of gastrectomy patients had it. Only one patient did not have cancer and underwent unnecessary gastrectomy. Further, on long-term follow-up (median 9 years), the true nutritional morbidity of total gastrectomy appears to be acceptable.

The importance of this study is that it supports the use of total gastrectomy for patients who have a family history of HDGC and have inherited CDH1 truncating mutation. However, we do not have long-term follow-up in the patients who refused surgery so we cannot compare them to patients who underwent surgery. A recent consensus conference suggests that endoscopic surveillance in expert centers is an alternative to total gastrectomy for the management of patients with CDH1 mutations [14]. However, a randomized trial between surgery and endoscopic follow-up is necessary to clearly determine if surgery is the best choice. This may be challenging because the number of these patients is limited. However, this issue may be overcome with the enrollment of an international clinical randomized trial or possibly a systematic review/meta-analysis. Since lymph node dissection failed to demonstrate any pathological lymph node metastases in patients with prophylactic gastrectomy, these data suggest that D-2 lymph node dissection is not necessary and may be discontinued. Further, this study suggests that endoscopic screening is not reliable for diagnosing the presence of signet ring cell cancer because there were 18 out of 22 patients who had a negative endoscopy and still had multifocal HDGC on final pathology. Patients who inherit a CDH1 mutation should undergo total gastrectomy, they should not wait until there is biopsy proof of cancer or symptoms develop. If patients have symptoms, our study indicates that they will have more advanced cancer and significantly reduced survival. The current results further support total gastrectomy because it suggests that operative complications are minimal and long-term quality of life is acceptable. Prophylactic gastrectomy is indicated for any patient with a family history of gastric cancer and a CDH1 mutation. These data were partially presented at the 2017 meeting of the Pacific Coast Surgical.

## Figures and Tables

**Table 1 cancers-14-00728-t001:** Presentation, number, sex, age, symptoms, and endoscopy results of patients with inherited CDH-1 mutations.

Presentation	Number	Women (%)	Mean Age (y) (Range)	Symptoms	Endoscopy + (%)
Symptomatic	8	63	40 (38–52)	GERD, Pain *n* = 3, weight loss, palpable mass, vomiting	8 (100%)
Asymptomatic	22	55	37 (27–71)	none	3 (14%)

**Table 2 cancers-14-00728-t002:** Presentation, number, pathology result, and survival at 7 years postoperatively.

**Presentation**	** *n* **	**Pathology Result**	**Dead with a Median Follow-Up of 7 Years (%)**
Symptomatic	8	T4N2M0 *n* = 1 T3N1M0 *n* = 5 T3N0M1 *n* = 1 T1N0M0 *n* = 1	5/8 (63%)
Asymptomatic	22	T1N0M0 *n* = 21 No tumor *n* = 1	1/22 (5%)

## Data Availability

The data presented in this study are available in article.

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
