# Peer review of "Surgery for Hereditary Diffuse Gastric Cancer: Long-Term Outcomes"

_cancers, 2022, doi:10.3390/cancers14030728_

Round 1
Reviewer 1 Report
OK
This manuscript is a resubmission of an earlier submission. The following is a list of the peer review reports and author responses from that submission.
Round 1
Reviewer 1 Report
This article by Forrester et al. describes how gene directed surgery affects long-term outcomes in patient with hereditary diffuse gastric cancer.
This reviewer recommends a major revision of the submitted manuscript before it can be considered for publiction. First of all, there are some major concerns regarding the scientific soundness and the presentation of results in the article.
The authors swicthes between presenting data as numbers (for instance 19 patients) and as written form (forty-threee). Please be consistent in the presentation.
Also, in the discussion at line 149, the authors use "We think...". This should be changed and the presentation of data from the study in the discussion should be more scientific. Likewise, the authors do not discuss limitations of the study. There is also no discussion regarding what novelty the study adds and future perspectives as well are not discussed. This should be added.
In addition, the authors use phrases such as "clearly documented" (line 114) which also sound non-academic. Please use other phrases.
When the authors discuss the why patients may decline surgery, they explain that open surgery is associated with more pain to the patients. The authors neglect discussing that patients are always informed of risks of any procedure including open gastrectomy, which may most likely affect the decision to decline surgery more. There is a higher mortality and complication rate to open gastrectomies.
Next, the introduction and the abstract are both not well-organized. It is hard to follow and understand the rationale behind the study design. The rationale behind the study as well as the hypothesis or expectations should be more clearly presented in the introduction.
Lastly, the subject is interesting but the authors fail to emphasize the importance of the study. This should be more clear when both presenting the paper (in the introduction) and when discussing the results.
Author Response
Authors’ Reply to the Review Report (Reviewer 1)
First, we want to thank reviewer 1 for his/her careful review of our manuscript “Gene Directed Surgery for Hereditary Diffuse Gastric Cancer: long-term outcomes.” We have carefully considered each of the reviewer’s comments and tried to edit the manuscript accordingly. Each of the changes made are marked by a red line in the left margin. We have revised the manuscript deleting the words for written numbers and used numbers as suggested. As suggested, we deleted the phrase “we think” and tried to present the data in a more scientific fashion. As suggested, we inserted a paragraph starting on line 158 to discuss limitations of the study. We discussed the novelty of the study line 167 and future directions on line 175 that indicates that lymph node dissection is no longer indicated for prophylactic patients. We removed “clearly documented” as suggested. We increased the paragraph of why patients may decline the surgery line 121. There is not a higher mortality rate of open gastrectomy compared to laparoscopic and we included two new references numbers 9 and 10 to support this contention. We changed the introduction and abstract for better organization and made the hypothesis and rationale of the study clearer. We emphasized the novelty line 136 and importance of the study on line 176. We hope that these revisions make the paper acceptable to reviewer 1. Again, we want to thank him/her for the insightful comments.
Reviewer 2 Report
This series is the result of a prospective study about prophylactic gastrectomy for hereditary diffuse gastric cancer. This manuscript is interesting, but I have some doubts/questions to be clarified.
1) The study started in 2004. When did the authors conclude it?
2) Why did the authors include symptomatic patients? In this case there is no propylactic surgery. At least a really gene directed surgery could be identified in these cases only if the CDH1 mutation implied an extension of surgical dissection.
3) It should be interesting to have some info of follow-up of patients declining surgery.
4) I congratulate with authors for their 0% morbidity rate, but they themselves admit that this surgery presents not so rare risks. Hence, I believe that the argumentation about the perception by patient of minimally invasive surgery implying a better compliance for prophylactic surgery needs a more clear discussion.
Author Response
First we want to thank reviewer 2 for the careful review of our manuscript. The study was started in 2004 and concluded in 2021 lines 58 and 59 in revised manuscript. We included symptomatic patients to be certain the reviewer realized that hereditary diffuse gastric cancer is a very bad prognostic disease such that prophylactic gastrectomy would be indicated. Since all the surgery was not done only on the presence of the CDH1 gene (not totally gene directed), we removed gene directed from the title. We agree that follow-up of the patients who declined surgery would be very interesting, but that was not written into the IRB
protocol so we do not have any follow-up of those patients. We tried to enhance the discussion about why patients may refuse surgery lines 121 to 127. Thank you for reviewing our work.
Round 2
Reviewer 1 Report
The manuscript has improved with the included changes. Yet, this reviewer would like to suggest some minor changes to the manuscript:
1). Do not start sentences with numbers. It disturbs the flow when reading the manuscript. For example at line 76 (first line in the results section); the sentence starts with "43..." Please write: "In total, 43..." Likewise at line 102, please try to start the sentences not with numbers but with words. The same issue can be seen at line 124.
2). There is something wrong with the sentence starting at line 173. I do not understand the sentence. Please revise.
3). Likewise, the sentence beginning at line 171 does not make sense to this reviewer. Please revise.
4). This reviewer would like the authors to add one or two sentences in the end of the manuscript, emphasizing that a randomized clinical trial is needed to confirm the findings of the present study but that this may be a challenging task as the number of patients are limited. However, this issue could be overcome with the enrollment of international clinical randomized trials. Furthermore, the existing literature on the matter could eventually be examined through a critical and systematical review and meta analysis. The evidence that is presented in this manuscript is still based on a cohort and thus do not represent the highest evidence in the hierarchy of evidence (such as RCTs and systematic reviews/meta analyses). Please includes a few lines addressing this issue.
5). Line 60: "Beginning August..." should be "Beginning in August..."
6). The authors write "the quality of life is acceptable" in the abstract, yet in the discussion section at line 188, the authors now write "the quality of life is excellent". Is the QoL excellent or just acceptable? Please be consistent.
7). The authors should place the sentence starting with "We prospectively studied the role of..." in the abstract under methods instead of under the introduction in the abstract.
I hope the authors will make the suggested changes and make the manuscript to go through control for grammatical errors (e.g. by English language control/check). If these changes are made and the language of the manuscript is checked, the manuscript will presumably be more suitable for publication.
Author Response
In, response to reviewer # 1, we thank you for your diligent review of the manuscript.
(1) We changed line 76, line 102, and line 124 as suggested.
(2) We tried to fix line 173 and it currently reads as follows: Because we had previously developed a method to serial section the entire resected stomach to find signet ring cell cancer2, we found a higher incidence of multifocal signet ring cell gastric cancer than expected. We expected only 70% of patients to have signet ring cell adenocarcinoma in the removed stomach1, but 95% of gastrectomy patients had it.
(3) We deleted the sentence line 171.
(4) Importantly, a randomized trial between surgery and endoscopic follow-up is necessary to clearly determine if surgery is the best choice. This may be challenging because the number of these patients is limited. However, this issue could be overcome with the enrollment of an international clinical randomized trial or possibly a systematic review/meta-analysis. This is placed in the last paragraph as suggested.
(5) We made the change on line 60 as suggested
(6) We made the QOL “acceptable” throughout.
(7) We moved the sentence to the methods as suggested.
(8) We checked the manuscript grammar with Microsoft Office and it received a grade of 95% with no suggested changes
We hope these changes make the manuscript acceptable in your opinion.
Reviewer 2 Report
My evaluation on this work is positive, but I find the revisions not so satisfying. Specifically:
1) "Since all the surgery was not done only on the presence of the CDH1 gene". What does it mean?
2) All the readers well know that "hereditary diffuse gastric
cancer is a very bad prognostic disease". The "comparison" (as presented in the Introduction) with symptomatic patients is unhelpful.
3) It is not so correct to specify that patients
declining surgery were lost to follow-up. In the study's protocol was there no recommendation for surveillance of these patients?
4) References should be improved: for example, the "new" reference #9 is a meta-analysis with 12 out of 13 trials reporting the use of laparoscopic subtotal gastrectomy. This is no so relevant for the comparison of lap vs op total gastrectomy as mentioned by the Authors.
5) Finally, the following sentence should be deleted: "The novelty of this study is that few other groups have done a similar study in patients with CDH1 mutations". In fact, this series should be presented as a real world analysis with some relevant aspects on an not so original issue with specific guidelines (that authors should include among their references: Blair VR, Lancet Oncol 2020).
Author Response
We want to thank reviewer # 2 for his or her comments and criticisms. We will try to answer each one.
(1) “Since all the surgery was not done only on the presence of the CDH1 gene.” This means that some of the patients had surgery because they had biopsy proven signet ring cell cancer. It was all the patients in the symptomatic group and 3 asymptomatic patients.
(2) I must disagree with the reviewer about the comparison with symptomatic patients. Our data shows that you cannot wait until the patient develops symptoms. You need to do the surgery based on the presence of a truncating CDH1 mutation. If you wait until they develop symptoms they will not do as well.
(3) The patients who declined surgery were followed at their home institution and treated there. Most of them were relatively young. They were advised to have repeat endoscopy every year and follow-up with a gastroenterologist. We do not have follow-up on the results of these endoscopies. So we cannot include the information. This has been revised on lines 97 and 98 as follows. Thirteen (30%) patients declined surgery and were advised to get yearly endoscopy with a gastroenterologist close to home. We do not have the results of these studies.
(4) I agree that reference #9 is a large review, but it does suggest that laparoscopy gastrectomy and open gastrectomy have similar complications. Most of the laparoscopic cases were not total gastrectomy as is required for HDGC. Laparoscopic is not better thar open except for less pain, but operative complications are similar. It remains in the revised manuscript lines 159-162.
(5) The recommended guidelines for HDGC management written in Lancet Oncology 2020 have been added as suggested. This is new reference 14. Lines 217-219
We have tried to answer your questions and make the revisions as you suggested. We hope that the manuscript is now acceptable for publication in Cancers.
Round 3
Reviewer 1 Report
This reviewer has read the manuscript again and believes the changes throughout the review process have considerably improved the manuscript. I thank the group of authors for making the relevant changes and for improving the manuscript especially in regards to the scientific soundness. This reviewer has no further comments and now find the manuscript suitable for publication.
Reviewer 2 Report
I appreciate the attempt to revise the manuscript by the Authors. I understand that they disagree with me, but in order to clarify their methods and results, they MUST clarify the difference between these sentences:
"All patients with inherited CDH-1 gene truncating mutations were prospectively studied on an IRB-approved clinical protocol" (Methods Section)
AND
"All the surgery was not done only on the presence of the CDH1 gene. This means that some of the patients had surgery because they had biopsy proven signet ring cell cancer". (Authors' Reply).
Authors well know that a SRC cancer is not necessarily an HDGC!